# Aromatic Hydrocarbon Removal by Novel Extremotolerant *Exophiala* and *Rhodotorula* Spp. from an Oil Polluted Site in Mexico

**DOI:** 10.3390/jof6030135

**Published:** 2020-08-14

**Authors:** Martín R. Ide-Pérez, Maikel Gilberto Fernández-López, Ayixon Sánchez-Reyes, Alfonso Leija, Ramón Alberto Batista-García, Jorge Luis Folch-Mallol, María del Rayo Sánchez-Carbente

**Affiliations:** 1Centro de Investigación en Biotecnología, Universidad Autónoma del Estado de Morelos, Cuernavaca 62209, Mexico; martin_ide@hotmail.com; 2Centro de Investigación en Dinámica Celular-Instituto de Investigaciones Básicas y Aplicadas, Universidad Autónoma del Estado de Morelos, Cuernavaca 62209, Mexico; fmaikel44@gmail.com (M.G.F.-L.); rabg@uaem.mx (R.A.B.-G.); 3Cátedras Conacyt-Instituto de Biotecnología, Universidad Nacional Autónoma de México, Cuernavaca 62209, Mexico; ayixon.sanchez@mail.ibt.unam.mx; 4Centro de Ciencias Genómicas, Universidad Nacional Autónoma de México, Cuernavaca 62209, Mexico; Alfonsoleijasalas@hotmail.com

**Keywords:** *Exophiala* sp., *Rhodotorula* sp., dimorphism, aromatic hydrocarbons, phenanthrene, benzo[*a*]pyrene, BTXs

## Abstract

Since Aromatic hydrocarbons are recalcitrant and toxic, strategies to remove them are needed. The aim of this work was to isolate fungi capable of using aromatic hydrocarbons as carbon sources. Two isolates from an oil polluted site in Mexico were identified through morphological and molecular markers as a novel *Rhodotorula* sp. and an *Exophiala* sp. Both strains were able to grow in a wide range of pH media, from 4 to 12, showing their optimal growth at alkaline pH’s and are both halotolerant. The *Exophiala* strain switched from hyphae to yeast morphotype in high salinity conditions. To the best of our knowledge, this is the first report of salt triggering dimorphism. The *Rhodotorula* strain, which is likely a new undescribed species, was capable of removing singled ringed aromatic compounds such as benzene, xylene, and toluene, but could not remove benzo[*a*] pyrene nor phenanthrene. Nevertheless, these hydrocarbons did not impair its growth. The *Exophiala* strain showed a different removal capacity. It could remove the polyaromatic hydrocarbons but performed poorly at removing toluene and xylene. Nevertheless, it still could grow well in the presence of the aromatic compounds. These strains could have a potential for aromatic compounds removal.

## 1. Introduction

Mexico is considered to be the fifth most megadiverse country in the world. This has been determined mainly by taking in account plants and animals, but microbes present in the many and varied ecosystems in the country has barely been explored. Mexico holds the fourth place in plant diversity [1], and this has been used previously as a marker to predict fungal biodiversity [2]. Unfortunately, large areas of Mexico have been polluted by petrol since Mexico is an oil-producing country [3]. Recently, the main source of pollution are fuel spills due to illegal activities to steal gasoline and diesel [3].

Diesel is composed approximately by 75% of paraffins, isoparaffins, cyloparaffins, and around 25% of aromatic hydrocarbons (AH), which include single ringed AH’s such as toluene, benzene, and xylenes and more complex polyAH’s (PAHs) such as Benzo [*a*] pyrene (BaP) and Phenantrene (Phe) [4]. The AH’s are recalcitrant and toxic molecules for most forms of life, so it is compelling to device methods to clean up areas in which these hydrocarbons prevail. Bioremediation is a good alternative for this purpose, since it is cheaper and friendlier with the environment than chemical or physical approaches [5].

Microbes have been shown to be good candidates to degrade many different kinds of xenobiotic molecules, ranging from pesticides and dyes to AH’s and even explosives as Trinitrotoluene (TNT) [6]. These processes are carried out mainly by oxidative, unspecific enzymes such as laccases, peroxidases, esterases, and cytochrome P450 proteins [7,8]. In some cases, hydrolases such as OPD (organophosphate degradation) enzymes are also involved [9]. These enzymes are able to oxidize a plethora of xenobiotic compounds and some organisms produce them and, along with a whole set of metabolic pathways, several monooxygenases and esterases can mineralize PAHs [10]. Bioremediation strategies have been focused on the use of bacteria and of white rot fungi since the latter produce oxidative enzymes that randomly oxidize and open the aromatic rings, which makes these molecules less toxic and available to other microorganisms [11]. However, in some cases, toxic products such as quinones are produced due to the oxidation of several xenobiotic compounds [12,13]. Hydrocarbon bioremediation by native yeasts is a still poorly explored field. *Rhodotorula* spp. yeasts are an exception since they have been widely studied in many aspects, from their function as probiotics in sea cucumbers (echinoderms that are considered to be a delicatessen and a source of pharmaceuticals) [14] and their ability to produce oils, carotenes, and enzymes [15,16] to their capacity to degrade hydrocarbons [17]. This genus also contains human pathogenic species [18,19]. However, few studies have focused on the characterization of *Rhodotorula* species to remove hydrocarbons in the presence of high concentrations of PHAs [20,21].

*Exophiala* spp. are opportunistic human pathogens [22]. A few reports have shown that *Exophiala* spp. were encountered in hydrocarbon polluted soils, but very little is known regarding *Exophiala* species in nature. They are considered “black yeasts,” which are a diverse group of fungi that belong to several orders, where many of them have been considered as extremophiles. *Exophiala* spp. are dimorphic yeasts that belong to the order *Chaetothyriales* [23,24,25]. The majority of the studies found in literature about this genus deal with medical issues, since most of the studied *Exophiala* species are human pathogens [24]. Thus, scarce studies are found of strains isolated from natural locations. This is also influenced by the difficulty to isolate these species from natural habitats. Penafreta-Boldú et al. [26] isolated a species belonging to this genus from an air filter designed to treat hydrocarbon-polluted gas, while Seyedmousavi et al. [27] could isolate an *Exophiala* species just after performing rounds of growth enriching the samples with heavy hydrocarbon derivatives such as creosote in a toluene-rich environment. Zhang et al. [28] also isolated a new *Exophiala* species from sub-Antarctic hydrocarbon polluted soil.

It has been proposed that there are two groups of *Exophiala* species in which one comprises human pathogens [29] and another that thrives in harsh conditions in nature able to tolerate extreme environments and to degrade hydrocarbons [26]. Data provided by Badali et al. [30] suggest that pathogenicity and extremophilic characters are not common to the same species and are found in ecologically-divergent groups. However, a recent study of several black-yeasts suggests the contrary: that similar traits are shared between pathogenic and biotechnologically potential species [31].

The aim of this work was to explore fungal species growing on petrol-polluted sites, which could have the capacity to remove aromatic hydrocarbons, since they are one of the most recalcitrant fractions of crude petrol.

## 2. Materials and Methods 

All the reagents were of an analytical grade and obtained from Fluka (Buchs, Switzerland) and Sigma-Aldrich (Saint-Louis, MO, USA).

### 2.1. Strain Isolation

The samples were collected at sites contaminated with oil and diesel in the town of Santa Isabel, Cunduacán, Tabasco, México (18°02′37.2″ N 93°04′18.1″ W) by a similar method, as described by Waksman [32]. One g of contaminated soil was resuspended in 9 mL of an isotonic solution (0.9% NaCl). Subsequently, serial dilutions (1:100, 1:1000) were made and 100 μL were placed on plates with potato dextrose agar (PDA) with 3% of diesel and 100 μg/mL of kanamycin and ampicillin to avoid bacterial growth. The cultures were incubated for 20 days at 28 °C or until colonies were present. Lastly, the isolates were purified to axenic cultures. From the isolated colonies, we made a second round of selection using mineral medium (CuSO_4_·H2O 7.8 mg/L, FeSO_4_·7H2O 18 mg/L, MgSO_4_·7H2O 500 mg/L, ZnSO_4_ 10 mg/L, KCl 50 mg/L, K_2_HPO_4_ 1 g/L, NH_4_NO_3_ 2 g/L, 3% bacteriological agar) with 3% of diesel, 10 ppm of Phe, and 10 ppm of BaP as carbon sources. The strains that were able to grow were cultured on plates with mineral medium and increasing concentrations (4%, 5%, and 6%) of diesel.

### 2.2. Identification of Isolated Strains

For molecular identification, DNA was extracted from the selected strains using organic phase purification [33]. To assess the integrity of the DNA, an electrophoresis was performed using a 1% agarose gel. PCR’s amplification of the ITS 2 region were performed using the oligonucleotides ITS3 (5′ GCA TCG ATG AAG AAC GCA GC 3′) and ITS4 (5′ TCC TCC GCT TAT TGA TAT GC 3′) with the following parameters: an initial denaturation cycle at 95 °C for 5 min, followed by 35 cycles at 95 °C for 45 s, 55 °C for 30 s, 72 °C for 1 min, and a final extension at 72 °C for 7 min. The amplicons were purified and sequenced. The sequences were used to performed a blast (http://www.ncbi.nlm.nhi.gov/BLAST) to obtain the sequences with higher homology. We selected a phylogenetic neighbor to use them in the construction of phylogenetic trees. To reconstruct the trees, we used BioNJ algorithm [34] K2P model for estimating genetic distances [35] and assessing the accuracy of the tree with 1000 bootstrap replicates. All steps were carried out in the program Seaview [36] version 3.5. The strains were given a strain identifier (BMH1013 and BMH1012) and were deposited in the collection of the Molecular Biology of Fungi Laboratory collection at the Autonomous University of Morelos. They are available upon request. The sequences of the amplified molecular markers for both strains are deposited in the GenBank (NCBI) with accession number MT268973 for BMH012 (*Rhodotorula* sp.) and MT268970 for BMH013 (*Exophiala* sp.).

### 2.3. Tolerance to pH, Temperature, and Salinity 

The tests were conducted by modification in methods from Henson, 1998 [37]. These parameters were assessed using the plate drip technique and 10 cfu of each isolate were inoculated on PDA plates in three biological replicas in triplicate each. For the pH tolerance tests, the media were adjusted using solutions of hydrochloric acid (1N) and sodium hydroxide (1N) until the desired pH (5–12) was obtained. To assess salinity tolerance, PDA plates with 1, 2, or 3 M of sodium chloride (NaCl) or without NaCl were inoculated with the selected isolates, as described above [38]. In the case of pH and salinity tolerance, the strains were incubated at 32 °C. For temperature tolerance assays, the strains were incubated at 20 °C, 32 °C, and 37 °C. Photographs were taken at the third day of all the conditions and, using the ImageJ program, we analyzed the pixel intensity to calculate the growth of the colonies.

For liquid fermentation, the growth rate was determined by counting the number of cells per mL in a Newbauer chamber every three days up to day 21. These experiments were also carried out in triplicate.

### 2.4. Determination of Hydrocarbon Removal by High Performance Liquid Chromatography (HPLC)

To determine the removal of hydrocarbons, we used a method modified from Xuezhu et al. [39]. Briefly, the strains were grown in 50 mL of mineral medium supplemented with a mixture containing: 0.1% benzene, 0.8% toluene, and 0.22% xylenes (ortho, meta, and paraxylenes) and 0.05% BaP and 0.05% Phe as carbon sources, incubated at 32 °C, at 200 rpm for 21 days. Every three days, homogeneous samples from the supernatant were taken to analyze the removal dynamics. To extract the hydrocarbons from the medium, a liquid-liquid extraction (70 mL of hexane per 30 mL of mineral medium) was performed, taking only the organic phase. The sample was dried and dissolved in a mixture of methanol: water (80:20). High performance liquid chromatography (HPLC) was used to determine the removal of hydrocarbons. A C18 column (0.25 mm in diameter per 30 cm in length) was used with a mobile phase pumping 0.8 mL/minute at a temperature of 30 °C. The removal percentage was determined according to the diminution of the peak area from chromatograms. Previously, standards were analyzed to determine the retention time for each hydrocarbon. The statistic test was one-way ANOVA (Tukey test) *p* < 0.05.

### 2.5. Evaluation of the Toxicity in Cucumber Seeds (Cucumis Sativus)

Germination tests were performed on a medium of water agar (1% agar agar). From the supernatants obtained on days 0, 6, 15, and 21, a liquid-liquid extraction (70 mL of hexane per 30 mL of mineral medium) was performed. The samples were dried until a volume of 6 mL was obtained. In addition, 2 mL of the hexane solution with the recovered hydrocarbons were mixed with 10 mL of water agar. Seven seeds of *C. sativus* were deposited in each Petri dish containing the water agar and the compounds were recovered in the extraction procedure. The seeds were incubated for 7 days at 25 °C. Lastly, the percentage of germinated seeds was determined and the size of the radicle was measured.

### 2.6. Statistical Analysis

All experiments were carried out in triplicate. The standard deviation was calculated for each experiment and depicted as a bar on the histogram. Analysis of variance (ANOVA) tests were applied to determine significant similarities or differences between measurements of the data. Lastly, a Tukey HSD (honest significance difference) post hoc analysis indicating the nature of the differences found in the ANOVAs was performed. All statistical calculations were performed using STATISTICA software version 7 STAT Soft Inc. Tulsa, Oklahoma, USA (data analysis software system). Levels of significance are always expressed as a *p*-value < 0.05.

## 3. Results and Discussion

### 3.1. Strain Isolation and Selection

Samples from polluted soils near a pipe conducting petrol derivatives in Santa Isabel, Cunduacán, Tabasco, were taken and the microbes in them were initially selected by growth in PDA medium supplemented with 3% diesel. Twelve different morphotypes grew under these conditions. Eleven out of the twelve isolates showed dome-like colony morphology, while strain BMH1013 was the only one showing mycelial growth. Microscopic observation of the isolates suggested that most of them were bacteria (based on the size and the absence of nuclei in the cells, see below) despite the addition of antibiotics to the selection plates. To test the ability of the isolates to use diesel as a sole carbon source, minimal media with serial upgrading concentrations of diesel (4%, 5%, and 6%) added with 10 ppm of Phe and BaP, were tested to isolate the most tolerant strains for further study. Although all twelve isolates could grow in PDA supplemented with 3% diesel, only strains BMH1012 (yeast like-phenotype) and BMH1013 (filamentous phenotype) were capable of growing at higher diesel concentrations (BMH1012 up to 4% diesel and BMH1013 up to 6% diesel). To further stress the robustness of the strains, they were grown in minimal media supplemented with 3% diesel and 10 ppm of BaP and 10 ppm of Phe as carbon sources. Both strains could grow under these conditions. Given the phenotypes encountered, only strains BMH1012 and BMH1013 were further characterized. We included a representative bacterial strain as a control to contrast the growth tolerance with BMH1012 and BMH1013 strains.

### 3.2. Identification of the Isolates

As stated above, BMH1012 formed creamy pink-colored colonies consistent with the “pink yeasts” in which many belong to the genus *Rhodotorula*. BMH1013 formed mycelial mats of a dark black color (Figure 1a,c).

Microscopic observation of the strains showed that strain BMH1012 showed yeast-like cells. The nuclei were clearly visible, which indicates its eukaryotic nature, while BMH1013 showed septate, annellidic hyphae, and conidia when grown in PDA medium (Figure 1b,d). These characteristics are consistent with the group of “black yeasts” [40]. The microscopic appearance was consistent with their macroscopic appearance.

Genomic DNA from both strains was extracted and subjected to amplification of the ITS2 region of the 28S ribosomal DNA coding sequences. Blast analysis of the sequences indicated that strain BMH1012 showed a 99.8% identity with strain NR_073296 of *Rhodotorula mucilaginosa*, while strain BMH1013 showed a 99.8% identity with strain KP959255.1 of *Exophiala heteromorpha* (Table 1). These data allowed us to retrieve collection type sequences of *Rhodotorula* (Figure 2a) and *Exophiala* (Figure 2b) species to perform phylogenetic reconstructions.

Robust phylogenetic trees were obtained, showing that the BMH1012 strain was clearly related to the *Rhodotorula* genus, although it places itself in a distinctively own branch, far from the rest of the *Rhodotorula* strains (with the exception of *Rhodotorula pacifica*, that has also its distinctive position in the phylogenetic tree, away from the bulk of the *Rhodotorula* species), which suggests that the isolate described in this case is a new undescribed species of the *Rhodotorula* genus (see for example, Reference [23]). BMH1013 strain is clearly related to *Exophiala heteromorpha*, although again, it is in a distinct branch of its closest relative, although always within the *Exophiala* genus (Figure 2), according to the branch support values between the two *E. heteromorpha* strains in the tree we propose that this is also a new species of *Exophiala*. Silva et al. [41] concluded that the sequence form rDNA internal transcribed spacer (ITS) regions is a reliable marker to classify these yeasts to the species level.

### 3.3. Growth in Different pH, Temperature, and Salinity Conditions

To further test the sturdiness of our isolates considering that polluted soils and waters might present harsh conditions, we decided to test the strains for growth in several stressful conditions.

Both strains showed the capacity to grow in a broad range of pH (Figure 3) and grew practically at the same rate from pH 5 to pH 12 with slight differences. The optimal for the *Rhodotorula* strain was 10 to 12 (Figure 3a), while, for the *Exophiala* strain, was 10 (Figure 3a). As a control, we grew a bacterial isolate from the same site to show that the pH and temperature tolerance was exclusive to the isolated yeasts.

The *Rhodotorula* sp. strain grew at 25 °C and showed an optimal growth at 32 °C, while *Exophiala* BMH1013 strain have an optimal growth temperature at 32 °C but not at 25 °C. The growth of the two strains was severely diminished at 37 °C. Most of the natural isolates are reported from cold climates, and even they have been found in clouds [42]. Therefore, generally, natural species seem not to be abundant in warm climates. However, most of the literature about *Exophiala* spp. deals with medical issues, since it is a human pathogen (consequently thriving at 36.5 °C). In conclusion, it seems this genus has a wide temperature range in which it can grow, but likely depends on the species and place of isolation.

Since oil spills occur frequently in salty environments, the growth of the strains in the presence of increasing concentrations of sodium chloride (NaCl) was tested. We found that both strains could grow in the presence of up to 1M NaCl, which is around the double of the concentration of salt in sea-water. The growth rate, however, was diminished with respect to the control without salt, which allows us to classify these strains as halotolerant (Table 2).

When performing the salt tolerance experiment, we noticed an interesting phenotype for strain BMH1013. Both its morphological and microscopic appearance were altered by the presence of NaCl (Figure 4). Macroscopic appearance changed from a mycelial mat to a yeast-like growth (Figure 4a,b, inner inserts). Microscopic observation was also different when comparing both conditions. In medium without salt, septate annellidic hyphae were observed as before (Figure 4a), while, in the presence of salt, more yeast-like cells were observed even though some thin hyphae structures were also seen. The conidiophore structure also suffered a morphological change (Figure 4b).

The results obtained in this study proved that the *Exophiala* strain could not grow beyond 37 °C and other reports have described the same condition (no growth beyond 40 °C) [43]. This characteristic could be useful to use this strain for hydrocarbon bioremediation since the temperature growth limitation would hamper infection of warm blood animals (including humans). Sav et al. [44] studied a big collection of *Exophiala* spp. (clinical strains as well as native isolates) with regard to several markers to compare both populations. Their conclusion was that there was inconsistency in the presence of virulence markers. Therefore, this suggests that *Exophiala* spp. are opportunistic microorganisms rather than pathogens [44].

Nevertheless, the strain described here was sturdy, since it could grow in a very wide range of pH almost at the same rate. Halotolerance was also an important trait, since it could grow in 0.5M NaCl, which is a concentration of salt that would impair growth of most land microorganisms or plants. An interesting phenomenon was observed when the *Exophiala* sp. was grown in the presence of high salt concentrations. This condition switched dimorphism so the strain grew more like a yeast than hyphae. Although dimorphic changes in fungi are usually triggered by stressful conditions of mainly pH or starvation [45,46,47], there is little information of fungal dimorphic switching in response to NaCl. Mehrabi et al. [48] showed that the HOG1 MAPK is involved in dimorphic switching in *Mycosphaerella graminicola* and Song et al. [49] also involved the HOG pathway in the dimorphic switch in *Metarhizium rileyi*. However, these microorganisms are not taxonomically related to the *Exophiala* genus. The HOG kinase is well characterized to be essential in the response to salt stress in many organisms [50]. To the best of our knowledge, this is the first report which directly demonstrates that fungal dimorphic change was triggered by high salt concentrations. Furthermore, there is a contrasting report about *Penicillium marneffei* that states: “Conversion of hyphae to yeast was rarely achieved at these salt concentrations” [51], which refers to the use of up to 4% NaCl. In contrast, it was found that a high NaCl concentration (around 5.8%) provoked the switch from hyphae to yeast morphology and still permitted growth for this *Exophiala* species.

### 3.4. Aromatic Hydrocarbon Removal by Strains BMH1012 and BMH1013

Since our goal was to isolate fungi with bioremediation capabilities, we tested the strains to remove a mixture of aromatic compounds resembling the aromatic fraction of diesel composition. Figure 5 shows the removal rate of different aromatic compounds by *Rhodotorula* strain BMH1012. The xylene standard showed two well-defined peaks in the HPLC and both isoforms were removed completely from the medium in 15 days (Figure 5c,d). Benzene and toluene removal was achieved at a slower rate than xylenes, but eventually were totally removed at the end of the experiment (Figure 5a,b). Interestingly, between days 15 to 18, a desorption event was detected for Benzene and Toluene, which were completely undetected at day 21 (Figure 5a,b). Neither BaP nor Phe were removed after 21 days of culturing. However, this did not impair growth of the *Rhodotorula* strain (Figure 5g).

In contrast, strain *Exophiala* sp. BMH1013 was able to remove efficiently the PAH’s. After nine days, only the 30% of BaP remained in the medium and, by the end of the experiment, an 80% removal was achieved (Figure 6f), while only a 35% of Phe was removed to the end of the experiment (Figure 6e). In contrast to the *Rhodotorula* strain, the single-ringed AH (with the exception of benzene) were poorly removed by *Exophiala* sp. BMH1013 (Figure 6a), especially toluene, which only showed a 25% removal. A 50% removal of xylenes was achieved from day 5 on, but no further removal was observed at the end of the experiment (Figure 6). Benzene had an interesting removal rate. In the first 12 days of the experiment, there was no change in concentration, while later on (from day 12 ahead), a rapid removal occurred until it was not detected (Figure 6a). *Exophiala* sp. BMH1013 presented a large lag phase that lasted six days and then growth was resumed at a good rate coinciding with benzene removal (Figure 6g).

Removal of hydrocarbons was achieved by both strains, but with different preferences. The *Rhodotorula* sp. removed single-ringed aromatic compounds (xylenes, toluene, benzene) but was completely unable to remove the more complex polyaromatic hydrocarbons (PAHs) Bap and Phe. While xylenes were almost completely removed by this strain in nine days, toluene and benzene showed a different kinetics. At day 12, they were almost non-detectable, but a desorption phenomenon took place at day 15 to be exhausted by day 21, which indicates that part of the initial removal was due to adsorption to the cell wall. When the other carbon sources were exhausted, these aromatic compounds (toluene and benzene) were released to the medium and used as a carbon source as suggested by diauxic phase of growth during days 15 to 18, there was an increase in the growth rate from day 18 to day 21 (Figure 5b). Although Bap and Phe were present at high concentrations during the whole experiment, they did not impair growth of *Rhodotorula* BMH1012, which indicates a high tolerance to these highly toxic compounds.

*Exophiala* BMH1013 in contrast could remove Bap and Phe (to around 20% and 38% of the initial concentration, respectively) but showed a poor removal of toluene (20%), and xylenes were only removed to about 50% of the initial concentration. The *Exophiala* strain started growth on day 9 (coinciding with a desorption of Phe), but it really entered the exponential phase of growth coinciding with benzene removal from day 15 to day 21 (Figure 6b). The presence of the other hydrocarbons such as xylenes did not seem to impede this strain from growing, which indicates a tolerance for the presence of these compounds. Likely, the hydrocarbons were removed in part by being adsorbed to the cell wall. When this structure gets saturated, the hydrocarbons stay in the medium.

Other works report the degradation of hydrocarbons by *Exophiala* and *Rhodotorula* spp. with high efficiency (ranging from 100% to 47%) (Appendix A). However, most of these reports use different timings media and temperature so it is difficult to compare directly their results with ours [28,52,53,54,55].

### 3.5. Ecotoxicity of the Treated Media in the Germination of C. Sativus

As a model to test toxicity after hydrocarbon removal from the liquid media and especially because some hydrocarbons remained present or are converted to more toxic forms, toxicity tests were conducted using a cucumber (*C. sativus*) germination seed assay [56]. Table 2 shows the percentages of germination of *C. sativus* seeds treated with extracts from supernatants of cultures from day 6, 15, and 21 of both fungi. The *Rhodotorula* strain BMH1012 was able to decrease toxicity of the medium only at day 6 of the treatment. Afterwards, germination was completely abolished when 15-days’ and 21-days’ supernatants were used (Table 3). However, the *Exophiala* strain showed a more promising potential, supernatants from days 6 and 15, allowed a 92.3% germination of the seeds, which indicates low toxicity of the remaining media. Unfortunately, and as in the case of the *Rhodotorula* strain, it seems that longer treatments generate toxic derivatives since only 76.9% of the seeds germinated when the 21-day supernatant was used.

Another parameter to determine toxicity was the length of the root of those seeds that germinated [57] (Figure 7a–k). A correlation with the germination percentage was observed. The higher the percentage of germination, the higher the length of the root, which indicates that not only germination was affected, but also the development of the seedlings.

Regardless of the removal of the hydrocarbons from the media, the remaining cultures were tested for toxicity at different times during the procedure. For the *Rhodotorula* strain, it was observed that only day 6 supernatants allowed germination of *C. sativus* seeds, but even less than the hexane control. The length of the seeds that could germinate was very short, which is significantly different from those in the hexane control. Since Bap and Phe were present at the initial concentration during all the experiments, the toxicity to the seeds cannot be accounted due to these compounds, since Day 6 supernatants still allowed germination. Rather, this behavior suggests the conversion of the initial single-ringed compound molecules into more toxic derivatives, since supernatants from days 15 and 21 completely abolished germination of the cucumber seeds (Figure 7 and Table 2).

However, extracts from supernatants from *Exophiala* allowed germination similarly to the hexane control on days 6 and 15, but not statistically different to the media with hydrocarbons and without treatment. In these days, xylenes were already removed to 50%, and Phe and Bap concentration was also diminished significantly. On day 21, a decrease in the germination was also observed. The root length in these experiments showed a concordance with the germination percentage (the higher the germination rate, the longer the root). Again, this suggests that derivative molecules generated by the fungal transformation are still toxic for the development of the root.

## 4. Conclusions

The main aim of this study was to isolate microorganisms with hydrocarbon-degrading abilities as a possible tool for bioremediation of oil polluted sites, which, in Mexico, are abundant and many times in extreme locations (sea water, salt marshes, warm tropical areas, etc.). In this work, new species of *Rhodotorula* and *Exophiala* were discovered and showed promising abilities to remove aromatic compounds from the media. They could grow well in the presence of aromatic hydrocarbons and they were tolerant to a wide range of pH’s and up to 1M NaCl, which makes them worth to study for bioremediation of polluted sites in which harsh conditions prevail. To the best of our knowledge, this is the first report in which a dimorphic switch is shown to be directly triggered by salt in fungi. The *Rhodotorula* strain could remove 100% of the single aromatic hydrocarbons, but not the polycyclic aromatic compounds (zero percent removal). On the contrary, the *Exophiala* strain could remove up to 80% of Benzo[*a*] pyrene and 60% of phenanthrene and only about 50% of the single-ringed aromatic compounds. Nonetheless, still some toxicity remained in the treated media, since an ecotoxicological test with cucumber seed revealed that germination and growth are impaired when in contact with the treated media after 21 days.

## Figures and Tables

**Figure 1 jof-06-00135-f001:**
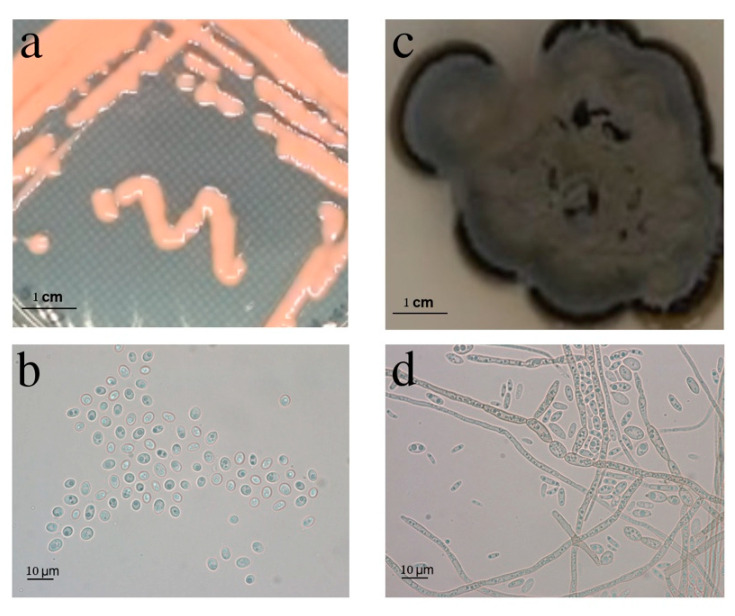
Macroscopic and microscopic morphology of the isolates BMH1012 and BMH1013. Macroscopic view of the strain BMH1012 (**a**) and BMH1013 (**c**) growing on a PDA Petri dish. Microscopic view of contrast phase of strain BMH1012 (**b**) and BMH1013 (**d**). Bar size: 10 μm.

**Figure 2 jof-06-00135-f002:**
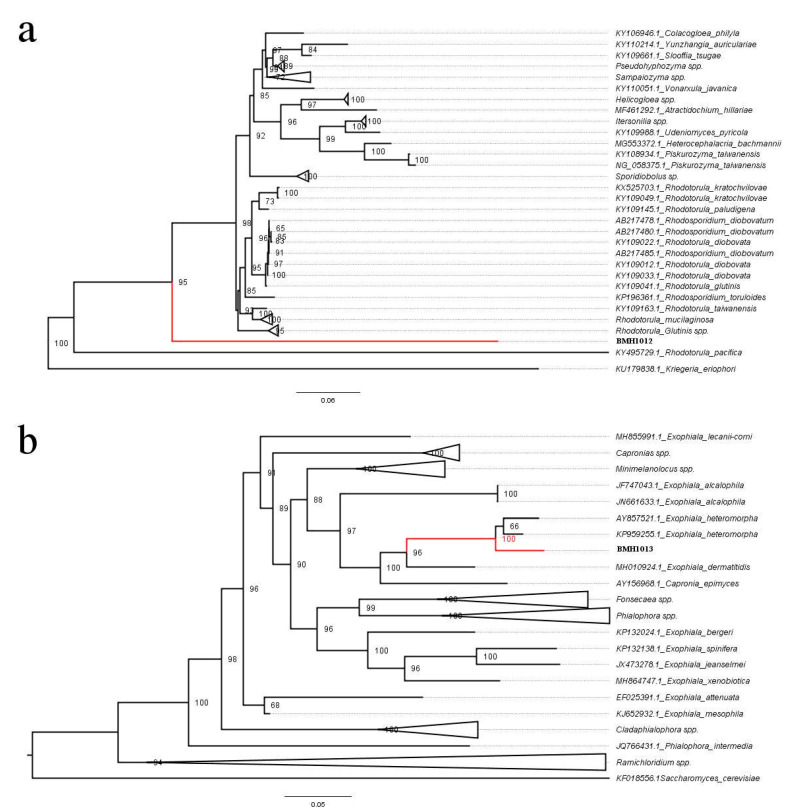
Evolutionary relationships of the Cunduacán isolated strains. (**a**) Phylogenetic relationships of strain BMH1012 with the *Rhodotorula* genus. and. (**b**) Phylogenetic relationships of strain BMH1013 with the *Exophiala* genus. The percentage indicated in the branches are the values obtained from the bootstrapped trees.

**Figure 3 jof-06-00135-f003:**
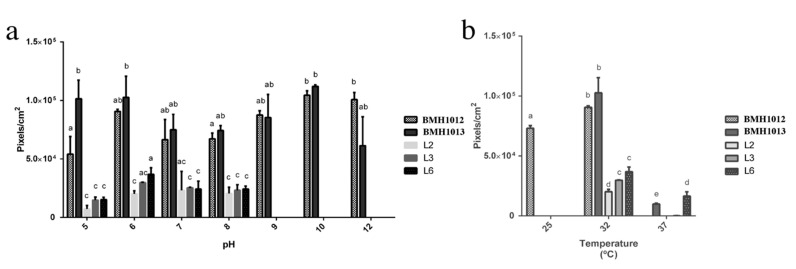
pH and temperature tolerance of the isolated strains. BMH1012 and BMH1013 are the yeasts isolates while L2, L3, and L6 are bacteria isolates. (**a**) Growth at different pH values of the indicated strains (**b**) Growth at different temperatures of the indicated strains. Letters indicate levels of statistical differences. ANOVA *p* value < 0.05.

**Figure 4 jof-06-00135-f004:**
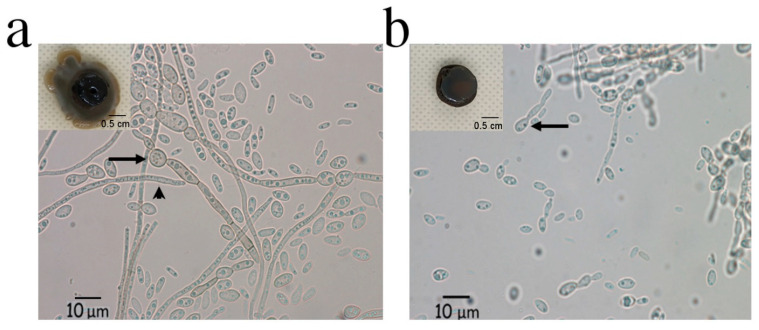
Macroscopic and microscopic morphological changes of strain BMH1013 induced by NaCl. The inner panels in (**a**) and (**b**) show the macroscopic view of the strains, in (**a**) cultures without NaCl; and (**b**) cultures with 0.5 M NaCl. The arrows show the change in the morphology of the conidiophore, while the arrowhead show the hyphae morphology.

**Figure 5 jof-06-00135-f005:**
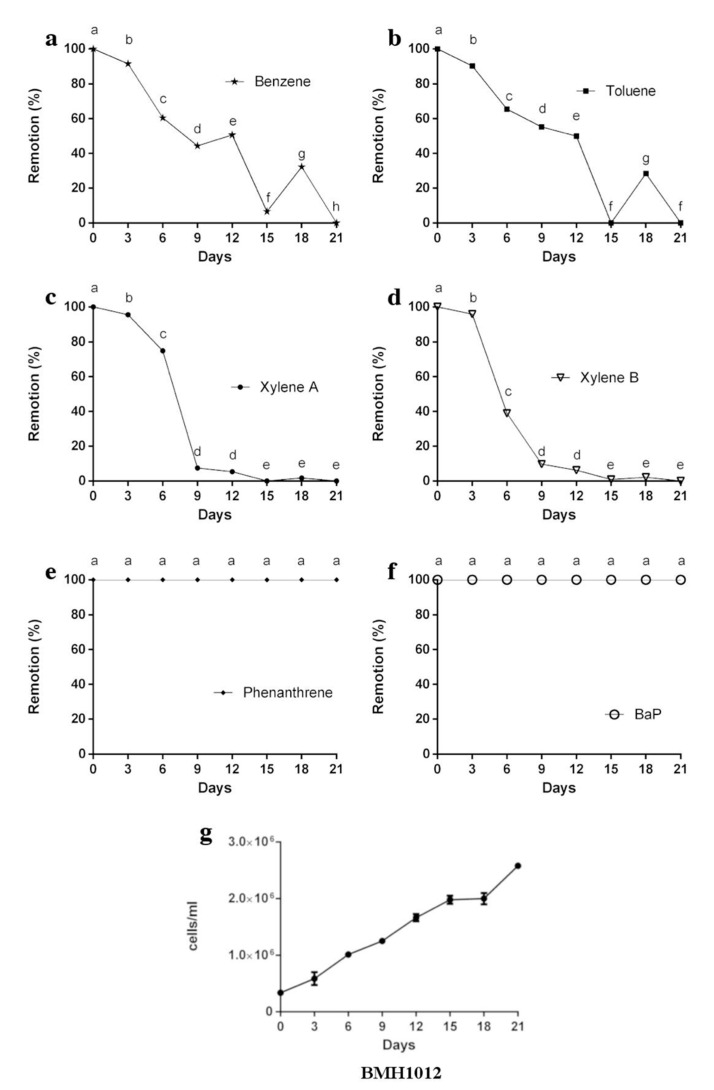
The remaining percentage of aromatic hydrocarbons removed by strain *Rhodotorula* BMH1012. (**a**) Benzene, (**b**) Toluene, (**c**) Xylene A, (**d**) Xylene B, (**e**) Phe, and (**f**) BaP removal after 21 days of fermentation. (**g**) Growth curve of *Rhodotorula* strain during fermentation with AH’s. Data is presented as the average amount of aromatic hydrocarbon remaining in the media (%) ± standard deviation (error bars). Letters indicate levels of significance and statistical differences. ANOVA *p*-value < 0.05.

**Figure 6 jof-06-00135-f006:**
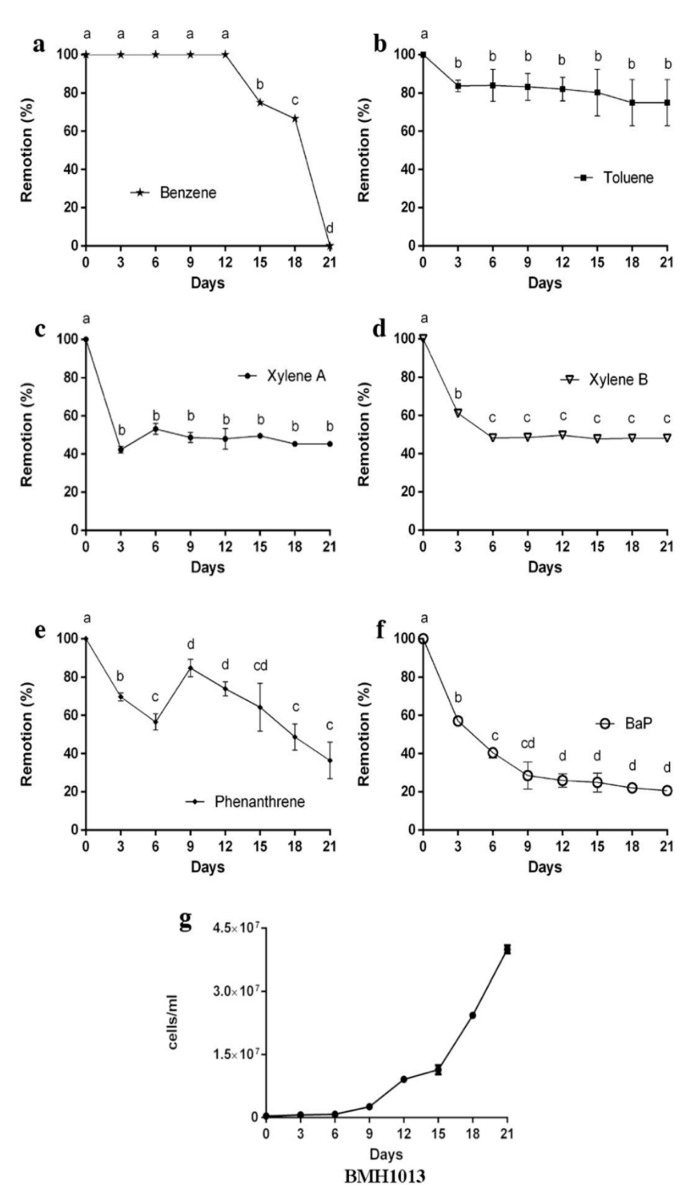
Percentage of aromatic hydrocarbons removed by strain *Exophiala* sp. (**a**) Benzene, (**b**) Toluene, (**c**) Xylene A, (**d**) Xylene B, (**e**) Phe, and (**f**) BaP removal after 21 days of fermentation. (**g**) Growth curve of *Exophiala* sp. strain during the fermentation with AH´s. Data is presented as the average amount of aromatic hydrocarbon remaining in the media (%) ± standard deviation (error bars). Letters indicate levels of significance statistically differences. ANOVA *p*-value < 0.05.

**Figure 7 jof-06-00135-f007:**
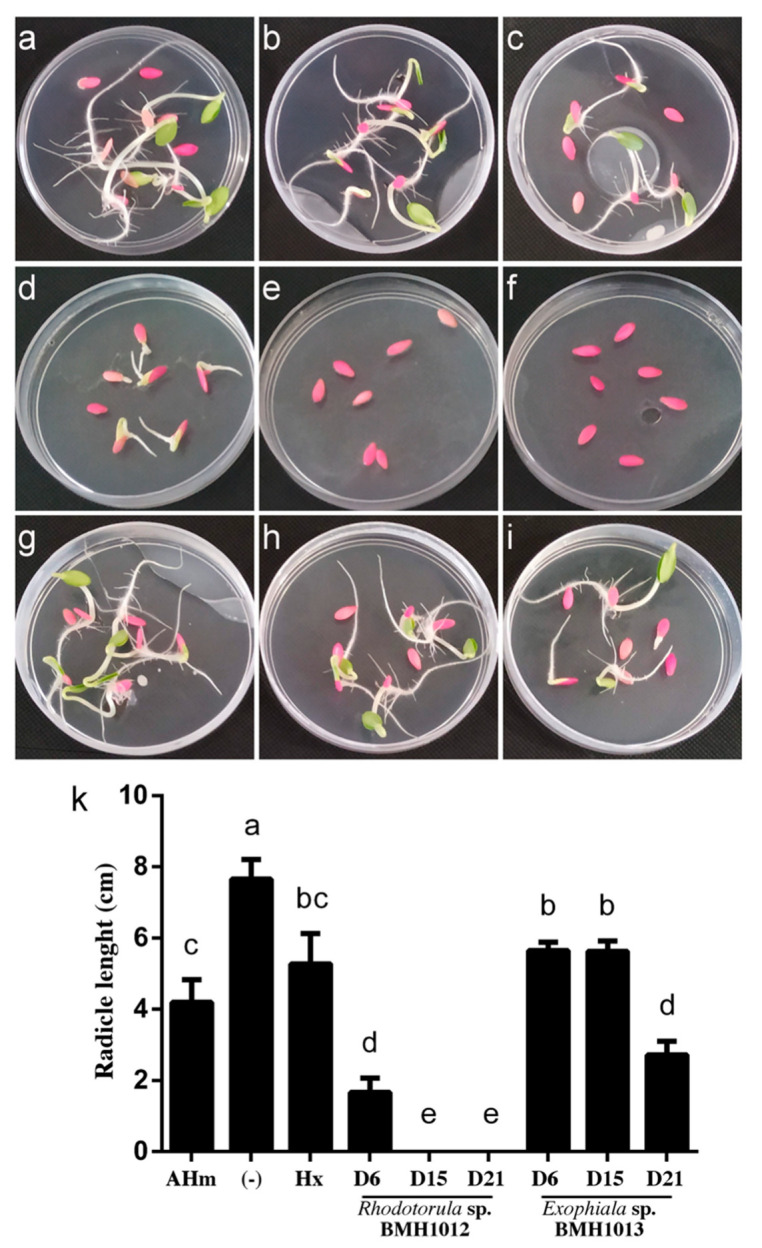
Effect on the germination and the radicle length of *C. sativus* seeds after treatment with extracts from supernatants of media with hydrocarbons treated by strains BMH1012 (D6, D15, and D21) and BMH1013 (D6, D15, and D21). (**a**) Crt-, agar with distilled water. (**b**) Agar with 20% hexane. (**c**) Control of agar with aromatic hydrocarbon mixture (AHm, see material and methods section), (**d**) Day 6, (**e**) D15, and (**f**) D21 supernatants from the culture of *Rhodotorula* sp. BMH1012 with AHm as a carbon source added to agar. (**g**) Day 6, (**h**) D15, and (**i**) D21 supernatants from the culture of *Exophiala* sp. BMH1013 with AHm as a carbon source added to agar. (**k**) Radicle length of seedlings growing on treatments as described above. Data is presented as the average amount of aromatic hydrocarbon remaining in the media (%) ± standard deviation (error bars). Letters indicate levels of significance and statistical differences. ANOVA *p*-value < 0.05.

**Table 1 jof-06-00135-t001:** Blast hits for both strains. The ITS2 region sequences obtained from the amplified PCR fragments were used to perform a BLAST in the NCBI server.

**Strain**	**Max Score**	**Total Score**	**Query Cover**	**E Value**	**% Identity**	**Accession**
*Rhodotorula mucilaginosa* strain JYC2617	665	665	0.99	0.0	98.93%	MN648703.1
*Rhodotorula* sp. strain SM03UFAM2	665	665	0.99	0.0	98.93%	MN268779.1
*Rhodotorula mucilaginosa* strain JYC529	665	665	0.99	0.0	98.93%	MN244371.1
*Rhodotorula mucilaginosa* YE-171	665	665	0.99	0.0	98.93%	LC486532.1
*Rhodotorula mucilaginosa* strain WUT167	665	665	0.99	0.0	98.93%	MN006818.1
*Rhodotorula* sp. strain DAMB1	665	665	0.99	0.0	98.93%	MK968443.1
*Rhodotorula mucilaginosa* strain IMUFRJ	665	665	0.99	0.0	98.93%	MK263185.1
*Rhodotorula mucilaginosa* JYC2513	665	665	099	0.0	98.93%	MK044010.1
**Strain**	**Max Score**	**Total Score**	**Query Cover**	**E Value**	**Per. Ident**	**Accession**
*Exophiala heteromorpha* strain IFRC 762	640	640	0.99	2 × 10^−179^	99.43%	KP959253.1
*Exophiala* sp. DAOM 216,391	640	640	0.99	2 × 10^−179^	99.43%	AF050267.1
*Exophiala heteromorpha* strain CBS 137222	634	634	0.99	8 × 10^−178^	99.15%	KJ522800.1
*Exophiala* sp. strain TC201	630	630	0.99	1 × 10^−176^	98.87%	MK465183.1
*Exophiala heteromorpha* isolate 21	630	630	0.99	1 × 10^−176^	99.15%	KC349856.1
*Exophiala heteromorpha* strain IFRC 686	623	623	0.99	2 × 10^−174^	98.58%	KP959252.1
*Exophiala heteromorpha* strain IFRC 761	617	617	0.98	8 × 10^−173^	98.57%	KP959251.1
*Exophiala heteromorpha* strain IFRC 813	612	612	0.95	4 × 10^−171^	99.41%	KP959255.1
*Exophiala heteromorpha* strain CBS 137223	612	612	0.99	4 × 10^−171^	98.02%	KJ522801.1

**Table 2 jof-06-00135-t002:** Fungal colony growth in the absence or presence of NaCl at different concentrations. The growth was measured as the diameter of the colonies.

Strain	Without NaCl	1M NaCl	2M NaCl
BMH1012	1.9 ± 0.3 cm	0.5 ± 0.1 cm	0 cm
BMH1013	1.7 ± 0.2 cm	0.5 ± 0.1 cm	0 cm
Ctr	1.2 ± 0.1 cm	0 cm	0 cm

Ctr: is one of the bacterial isolates.

**Table 3 jof-06-00135-t003:** Effect in germination of cucumber seeds of the supernatants of the media with AH´s mixture treated with BMH1012 and BMH1013 strains.

Strains	Treatment	Germination (%)
Controls	(−)	100 ± 0.58
	Hexane	92.3 ± 0.0
	AHm	61.5 ± 0.58
BMH1012	D6	84.6 ± 0.58
	D15	0.0 ± 0.0
	D21	0.0 ± 0.0
BMH1013	D6	92.3 ± 0.0
	D15	92.3 ± 0.0
	D21	76.09 ± 0.0

Control (−), agar-water media, control hexane, agar-water media supplemented with 20% hexane (the solvent used for the AH´s extraction), control (+) agar-water supplemented with AH’s mixture dissolved in hexane (AHm). D6, D15, and D21 are the days where the samples from the fungal cultures of media with AH´s mixture (as carbon sources) were collected and extracted with hexane.

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
