# Peer review of "Aromatic Hydrocarbon Removal by Novel Extremotolerant Exophiala and Rhodotorula Spp. from an Oil Polluted Site in Mexico"

_jof, 2020, doi:10.3390/jof6030135_

Round 1

Reviewer 1 Report

This paper shows the ability of two types of isolated microbes for the removal of aromatic and polycyclic hydrocarbons. The paper is well written with informative abstract and introduction. just needs some changes.

Introduction: the mechanism of removal by selected microbes should be explained in detail.

Sections 2.1, 2.3, and 2.4 need to be cited.

Lines 242-246: these lines are limitations of this study? It means the isolated microbes are tolerant to high salt concentrations. when does the tolerance finish? Did you continue to the extreme salt concentration? 

As figure 5 and 6 are the main results of this study, so the comparison of these results with similar genome microbes in published studies are necessary. It shows the difference and capability of their removal. I suggest doing a comparison in the table not writing text to show the experimental condition in similar studies as well.

Figure 6 (a): what happens after day 18? Does the value become zero? is there any explanation for this result?

The discussion section is not necessary. It can be merged with the results. For instance. lines 373-396 can be written after figures 5 and 6. The rest of discussion also can be written after related figures and tables.

What about the optimum condition for the removal of PAHs by selected strains? did the authors study? 

The conclusion is too short and can be extended to more results such as removal rate and germination rate in different conditions.

Some recent and related studies can be added to the reference list. (2018-2020).

Author Response

Reviewer 1

In black, the reviewer’s queries. The blue text are our answers; in green the text added to the manuscript:

This paper shows the ability of two types of isolated microbes for the removal of aromatic and polycyclic hydrocarbons. The paper is well written with informative abstract and introduction. just needs some changes.

Introduction: the mechanism of removal by selected microbes should be explained in detail.

Lines 54-58: a paragraph broadly explaining the degradation mechanisms has been added in the Introduction section with its corresponding references; to avoid repeating information in lines 60 and 64 “…such as laccases and peroxidases… this paragraph was modified and an explanation about why these enzymes transform the compounds to easier available sources has been added:

(Now lines 54-58) These processes are carried out mainly by oxidative, unspecific enzymes such as laccases, peroxidases, esterases and cytochrome P450 proteins7,8. In some cases, hydrolases such as OPD (organophosphate degradation) enzymes are also involved.9 These enzymes are able to oxidize a plethora of xenobiotic compounds and, along with a whole set of different metabolic pathways, can mineralize xenobiotic compounds10

Bioremediation strategies have been focused in the use of bacteria and of white rot fungi, since the latter produce oxidative enzymes that randomly oxidize and open the aromatic rings, making these molecules less toxic and available to other microorganisms11.

Sections 2.1, 2.3, and 2.4 need to be cited.

Lines 99-100 (section 2.1): A citation has been introduced as requested so now it reads:

Fungi were collected by a similar method as described by Waksman32

Waksman SA, 1922 A method of counting the number of fungi in the soil. J Bacteriol 7:339–341

2.3 The citation has been introduced in line 129:

The tests were conducted by modification in methods from Henson, 199837

  1. Benson, 1998. Microbiological Applications: A Laboratory Manual in General Microbiology. Mac-Graw-Hill 7th Edition.

2.4 Line 144: A citation was included:

Xuezhu, et al. 2016. Biodegradation of Mixed PAHs by PAH-Degrading Endophytic Bacteria. International Journal of Environmental Research and Public Health 13, 805.

So now the text reads:

To determine the removal of hydrocarbons, we used a method modified from Xuezhu, et al.39 Briefly,…

Lines 242-246: these lines are limitations of this study? It means the isolated microbes are tolerant to high salt concentrations. when does the tolerance finish? Did you continue to the extreme salt concentration? 

On the contrary, these organisms were classified as halotolerant, which is a treat that few microorganisms possess. This means that they can grow, for example in marshes close to the coast lines, where salt concentrations are around the ones endured by our isolates. The average for seawater is 0.5M NaCl. It is also true that very few places on Earth have higher salt concentrations (the Dead Sea, for example). So, the isolates described here are robust fungi that could degrade hydrocarbons in the sea or other salty areas. Since no growth was observed at 2M NaCl, we did not further test higher salt concentrations.

As figure 5 and 6 are the main results of this study, so the comparison of these results with similar genome microbes in published studies are necessary. It shows the difference and capability of their removal. I suggest doing a comparison in the table not writing text to show the experimental condition in similar studies as well.

After an exhaustive search in literature, the following table was achieved. However, the reader should be careful because the media, conditions of growth, temperatures and timings in which these hydrocarbons were removed, varied significantly in each work and this makes it difficult to compare with our findings, so we decided to include the Table in supplementary materials, and a write a brief text in the discussion section (now merged with the Results section, in lines 364-367) referring to the Table. Other hydrocarbons such as anthracene or pyrene have also been reported by these genera. The curious reader then can access the references for details.

Strain

Compound/concentration

% removal

Reference

Exophiala sp. UTMC 5043

Phenanthrene

100 ppm

89

Akbarzadeh Farahnaz 2018

Exophiala macquariensis

Toluene

80 ppm

50

Zhang 2019

Exophiala. oligosperma

Toluene

50 ppm

100

Estevez et al, 2005

Rhodotorula mucilaginosa A29

Toluene

150 ppm

88.29

Hesham et al 2018

Rhodotorula diobovatum

Toluene

150 ppm

85.3

Hesham et al 2018

So, now, in lines 364-367 it reads:

Other works report the degradation of hydrocarbons by Exophiala and Rhodotorula spp. with high efficiency also (ranging from 100 to 47 %) (Supplementary Table S1). However, most of these reports use different media, timings, and temperatures so it is difficult to compare directly their results with ours52,28,53,54,55

Figure 6 (a): what happens after day 18? Does the value become zero? is there any explanation for this result?

This experiment was followed for 21 days. The only hydrocarbon that was completely removed to non-detectable levels was Benzene at day 21. After 21 days the experiment was not followed further since it looked like the system had already entered a stationary removal pattern for the other compounds (except maybe for phenanthrene, but this was not measured further). Probably the hydrocarbons are removed greatly by being adsorbed to the cell wall, when this structure gets saturated, the hydrocarbons stay in the medium. So, we added a sentence in that section:

Lines 362-363

…tolerance to the presence of these compounds. Probably the hydrocarbons were removed in part by being adsorbed to the cell wall, when this structure gets saturated, the hydrocarbons stay in the medium.

The discussion section is not necessary. It can be merged with the results. For instance. lines 373-396 can be written after figures 5 and 6. The rest of discussion also can be written after related figures and tables.

The Results and Discussion sections have been merged as suggested, each paragraph is now placed after the description of the results close to the Figures or Tables

What about the optimum condition for the removal of PAHs by selected strains? did the authors study? 

Unfortunately, we have a limitation of temperature-controlled shakers, and a lot of students in the group, so it was not possible to test many conditions. Here we report the removal of hydrocarbons in the optimal conditions of growth for each strain

The conclusion is too short and can be extended to more results such as removal rate and germination rate in different conditions.

The main findings of the study are now in the Conclusion section

Some recent and related studies can be added to the reference list. (2018-2020).

Yes, many recent references have been added along the text (also a very old one from 1922, regarding the way to isolate fungi from soils, such it is). See for example references 5, 6, 7, 9, 11, 20, 21…28, 31, 51, 53, etc.

Reviewer 2 Report

In this work Sanchez-Carbente and coworkers carried out an exhaustive study about the capability of the Rhodotorula strain to remove some aromatic compounds.

The topic is significant, and the article is clear and comprehensible even for non-specialists. For these reasons we consider that it deserves to be published in the Journal of Fungi with minor revisions.

More details should be given in the header of Table 1.

ml should be replaced by mL along the manuscript

In table 1, the format of Query values (0,99) and E values (2E-179, 8E-178, etc.) should be corrected.

The quality of Figure 2 should be improved

On page 8 "see for example19" it should read: "see for example Reference 19"

On page 15, the authors state: "Another parameter to determine toxicity was the length of the root of those seeds that germinated"

A reference about this interesting procedure should be added.

On page 16, the authors state: "The results obtained in this study proved that the Exophiala strain could not grow beyond 37 °C and other reports have described the same condition (no growth beyond 40 °C)"

The question naturally raised is: What is the maximum temperature at which Exophiala still grown ?

Author Response

Reviewer 2

In black, the reviewer’s queries. The blue text are our answers; in green the text added to the manuscript:

In this work Sanchez-Carbente and coworkers carried out an exhaustive study about the capability of the Rhodotorula strain to remove some aromatic compounds.

The topic is significant, and the article is clear and comprehensible even for non-specialists. For these reasons we consider that it deserves to be published in the Journal of Fungi with minor revisions.

More details should be given in the header of Table 1.

Lines 219-220: Now the header reads:

Table 1. Blast hits for both strains. The ITS2 region sequences obtained from the amplified PCR fragments were used to perform a BLAST in the NCBI server

ml should be replaced by mL along the manuscript

Thank you for the observation, it has been corrected all along the manuscript. Now all the quantities are expressed as mL

In table 1, the format of Query values (0,99) and E values (2E-179, 8E-178, etc.) should be corrected.

Thank you again for the observation, all the comas in the Query values have been substituted by periods and the E-values corrected by formatting the value with a super index.

The quality of Figure 2 should be improved

On page 8 "see for example19" it should read: "see for example Reference 19"

This has been added, now in line 228 it reads

described here is a new undescribed species of the Rhodotorula genus (see for example reference 23).

On page 15, the authors state: "Another parameter to determine toxicity was the length of the root of those seeds that germinated"

A reference about this interesting procedure should be added.

Thank you for this suggestion. The following reference has been added:

Sivaram, et al., 2019. Phytoremediation efficacy assessment of polycyclic aromatic hydrocarbons contaminated soils using garden pea (Pisum sativum) and earthworms (Eisenia fetida). Chemosphere 229; 227-235

On page 16, the authors state: "The results obtained in this study proved that the Exophiala strain could not grow beyond 37 °C and other reports have described the same condition (no growth beyond 40 °C)"

The question naturally raised is: What is the maximum temperature at which Exophiala still grown ?

We did not try beyond 37 ºC, in which this particular strain does not grow any more. There is a lot of variance between different Exophiala isolates. Most of the natural isolates are reported from cold climates, even they have been found in clouds (Das Sarma, et al., 2020. Earth's Stratosphere and Microbial Life. Curr. Issues Mol. Biol. Vol. 38: 197-244). So generally, the natural species are not very fond of warm climates. However, most of the literature about Exophiala spp. deals with medical issues, since it is a human pathogen (consequently thriving at 36.5 ºC.). In conclusion, it seems this genus has a wide temperature range in which can grow, but probably depends on the species and place of isolation. In our case it was isolated from a warm tropical site, so it is not surprising that it can grow at 32 ºC

This has been included in the manuscript, so in lines 254-259 you can read:

… Most of the natural isolates are reported from cold climates, even they have been found in clouds42 ,so generally, natural species seem not to be abundant in warm climates. However, most of the literature about Exophiala spp. deals with medical issues, since it is a human pathogen (consequently thriving at 36.5 º C.). In conclusion, it seems this genus has a wide temperature range in which it can grow, but probably depends on the species and place of isolation.
